# Effectiveness of targeting the health promotion settings for non-communicable disease control in low/middle-income countries: systematic review protocol

Gursimer Jeet,[1] Jarnail Singh Thakur,[1] Shankar Prinja,[1] Meenu Singh,[2] Ronika Paika,[1] Kunjan Kunjan,[1] Priya Dhadwal[1]

¹School of Public Health, Postgraduate Institute of Medical Education and Research, Chandigarh, India
²Advanced Paediatric Centre, Postgraduate Institute of Medical Education and Research, Chandigarh, India

**Correspondence to**
Dr Gursimer Jeet;
gsj_2008@yahoo.com

## ABSTRACT

**Introduction** Settings-based approaches to health promotion, involving holistic and multidisciplinary methods, which integrate action across risk factors are important. Major advantage of focusing on these settings is the continuous and intensive contact with the participant. Despite the apparent advantages of addressing non-communicable diseases (NCDs) using targeted interventions for several developed country settings, a relative lack of evidence of effectiveness of such interventions in low/middle-income countries has led to poor allocation of resources towards these interventions. The focus is therefore on the settings rather than any one condition, and we therefore expect the findings to generalise to NCD prevention and control efforts. We intend to estimate the effectiveness of targeted interventions in low/middle-income countries.

**Methods and analysis** We will search PubMed, Excerpta Medica Database, OVID, WHO Library and The Cochrane Library from the year 2000 to March 2018 without language restrictions. Study designs to be included will be randomised controlled trials. The primary outcome of effectiveness will be the percentage change in population having different behavioural risk factors. Subgroup analyses will be performed, and sensitivity analyses will be conducted to assess the robustness of the findings.

**Ethics and dissemination** No ethical issues are foreseen. The Institute Ethics Committee of the Post Graduate Institute of Medical Education and Research approved the doctoral research protocol under which this review is being done. Dissemination will be done by submitting scientific articles to academic peer-reviewed journals. We will present the results at relevant conferences and meetings.

**Study design** Systematic review.

**PROSPERO registration number** CRD42016042647; Pre-results.

## BACKGROUND

Prevalence of risk factors of non-communicable diseases (NCDs) among populations in low/middle-income countries is high.[1 2] In

### Strengths and limitations of this study

► The review will contribute towards evidence synthesis on effectiveness of settings-based approach for reducing burden of non-communicable disease risk factors in resource-constrained settings.
► This will be a first of its kind review resulting in synthesis of evidence from randomised controlled trials on targeted settings in low/middle-income countries.
► The synthesis of evidence can probably act as an advocacy tool for better resource allocation towards settings-based approach.
► Due to resource constraints, focus is limited to low/middle-income countries. A comprehensive review will require evidence generation through comparison of the effects of the design in developed and developing settings.

2012, NCDs contributed towards 68% of total deaths in the world with 80% of deaths in low/middle-income countries.[2] The figure is more alarming than it appears as 52% of the NCD deaths that occurred were premature, that is, among people aged 30–70 years.[2] In the year 2012, 48% of all deaths in Southeast Asia were premature.[3] As the four major NCDs, that is, cardiovascular diseases, diabetes, cancer and stroke share four modifiable risk factors (tobacco, alcohol, diet and physical activity), the prevention component is very crucial for this group of diseases. There is a global call to develop interventions that address the risk factors using different health promotion approaches.[4 5] There is good evidence that health promotion interventions focusing on changing lifestyle behaviours are more effective if conducted by targeting settings[6] rather than population-based approaches alone.

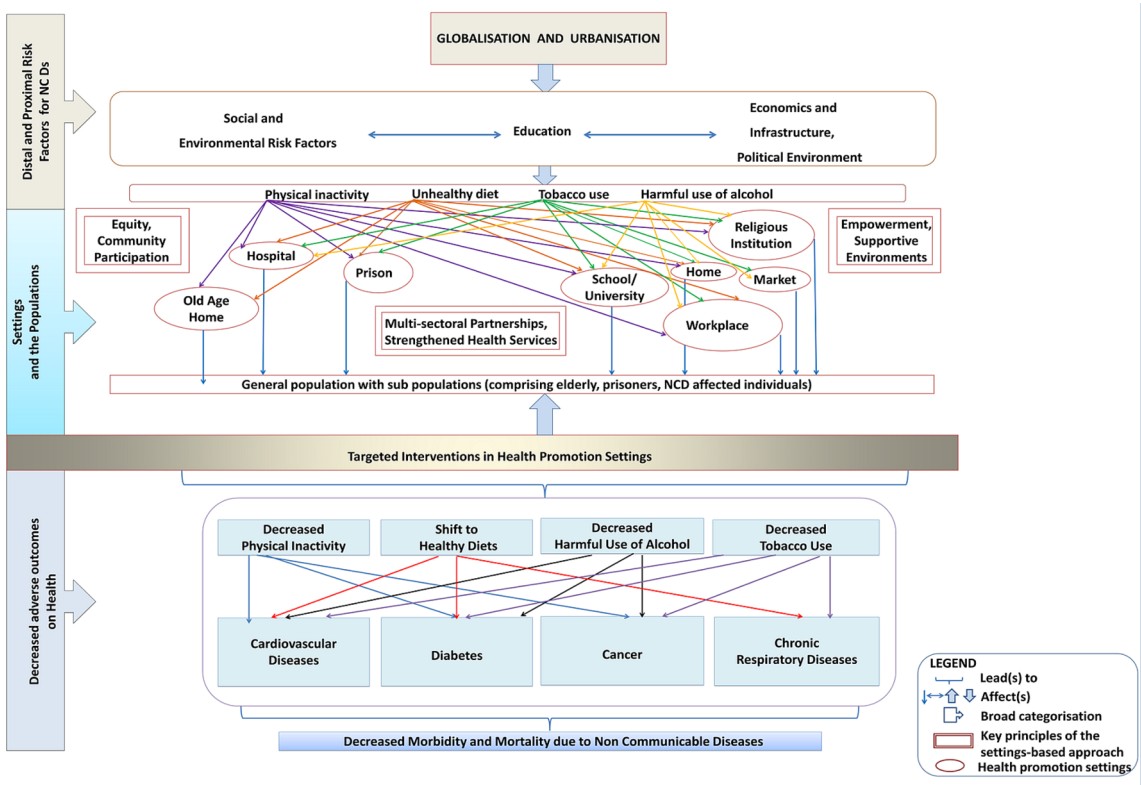

**Figure 1** Proposed framework of impact of targeting the health promotion settings for non-communicable disease (NCD) control in low/middle-income countries.

Therefore, settings-based approaches to health promotion, involving holistic and multidisciplinary methods, which integrate action across risk factors have become important.[4 7] Major advantage of focusing on these settings is the continuous as well as intensive contact with the participant. It is also indicated that support from staff, suitable infrastructure, physical environment and health promoting policies have potential to positively influence the health of a person especially in these kinds of settings.[8] In order to decrease the burden of proximal and distal determinants of NCDs which has increased due to globalisation and urbanisation, settings-based approaches may be implemented in a complete and inclusive manner by targeting singular or multiple settings simultaneously (figure 1). These have to be guided by principles of equity, supportive environments, empowerment, community participation and multisectoral partnerships with strengthened health services to achieve maximum benefit in terms of disease prevention.[9] The framework given as figure 1 helps to understand how focusing on risk factors prevalent in the 'settings' can help in finding a solution to NCDs at a larger scale. Efforts can be channelled through general population or subpopulations. Additive or multiplicative effect of working through settings can be expected, as some settings are very closely related like workplaces and homes, schools and homes. A change in behaviour patterns and in turn risk profile with respect to NCDs can be expected to percolate from one setting to the other. This approach is likely to alter behaviour patterns in the form of decreased risk factor

levels and consequently a shift in the incidence and prevalence of cardiovascular diseases, diabetes, cancers and chronic respiratory diseases. Finally, the rate of increase in burden of morbidity and mortality due to NCDs can be slowed down by using targeted interventions in various settings.

In this review, strategy of targeted interventions is being evaluated for control of NCD risk factors under different health promoting settings mentioned in the framework. The settings approach has proven to be effective in control of several other risk factors or diseases such as HIV,[10–12] immunisation,[13] etc.

Despite the apparent advantages of addressing NCDs using targeted interventions for several developed settings,[14 15] a relative lack of evidence of effectiveness of such interventions in low/middle-income countries[16] has led to poor allocation of resources towards these interventions. The systematic review aims to assess existing evidence on effectiveness to guide health promoting sites to accept, adopt and correctly deliver targeted interventions for NCDs.

## METHODS

In reporting the protocol for this review, Preferred Reporting Items for Systematic reviews and Meta-Analyses Protocols checklist has been adhered to and is provided as a supplementary file along with the main manuscript (online supplementary file 1).[17]

## Population

The populations of interest will be the different stakeholders involved in design, implementation and sustainability of targeted NCD control interventions in different settings from low/middle-income countries.

## Intervention details

Interventions here refer to use of 'settings-based approach for delivery of health promotion interventions'. Settings here are defined as, 'The place or social context in which people engage in daily activities in which environmental, organisational and personal factors interact to affect health and wellbeing.'[18] They can be identified as having physical boundaries, a range of people with defined roles and an organisational structure. Schools and workplaces have been popular since long as a setting for implementation of such interventions.[6] However, the definition and scope of settings have widened since last few decades with acceptance of a wide range of health promoting settings, such as cities, villages, municipalities/communities, workplaces, markets, home, hospitals, islands, prisons and old age homes.[9]

For school-based targeted intervention studies in primary, middle, high secondary and central schools (a selective secondary education school in some countries or the schools that are affiliated to central government in some countries) irrespective of their being private, public or aided (school receiving a part of its maintenance costs from public funds but retaining control over appointments and religious instruction) will be considered for inclusion. The inclusion criterion for mean age of students will be age between 5 and 18 years. For college, the trials recruiting students from institutions with bachelor, master's courses or other professional recognised courses will be included in the study. For schools and colleges, participants can be teaching staff and non-teaching staff, in which case trials may be classified as workplace settings. Participants may be individuals, classes, groups or whole institution.

For work site interventions, we will include trials set in government, non-government, aided institutions, organised or unorganised sectors set in low/middle-income countries. These may be involved in implementation of NCD control interventions or programmes by the means of encouragement, accreditation or enforcement. Participants may be individuals, group of workers or whole institute.

## Comparator

Routine care or enhanced routine care.

## Inclusion and exclusion criteria

Since we are focusing on the settings, cluster randomised controlled trials (cRCTs) will be the most eligible study design for inclusion. However, individual RCTs will be included if followed in any study. The controlled trials, quasi-experimental study designs may be the feasible design for intervention implementation, but in light of their inherent weaknesses such as lack of control area or lack of random assignment; internal validity gets reduced and causal inference becomes difficult.[19] Therefore, these study designs will be excluded from the review.

Only those interventions are selected that are disseminated using settings-based approach. We will include studies comparing healthy settings-based strategies with target of promoting behaviours that are healthier when it comes to undertaking physical activity, patterns with respect to diet, use of tobacco, consumption of alcohol against usual practice or enhanced usual care. A protocol detailing methodology of this review was registered at International Prospective Register of Systematic Reviews.[20] Strategies of intervention may include multidisciplinary expert training groups, education and training part of strategy may involve discussions, group tasks, regular tests, etc, followed by expert group recommendations for quality improvement, reminder letters and follow-up assessments for overall reduction in risk factors.[21] The interventions under scope of this process can be singular or with multiple components. These can be directed at particularly individuals, groups or complete institutions.[21 22] The model of delivery of interventions can belong to (but is not limited to) change in organisation policies or practices, enhanced family support, social networking, change in physical environment or routine practices.

Types of outcome measures: Selection of outcomes for the review is guided by nature of interventions under study which focus primarily on promoting the healthy behaviours. We have selected change in participant behaviour and a consequent shift in their physical (eg, weight, blood pressure), biochemical or cognitive parameters as primary outcomes as these are directly related to participant's health status. Additionally, as secondary outcomes, we will report rate of use of services or cost/cost-effectiveness of implementing interventions.

### Primary outcomes

1. Self- reported change in behavioural parameters.
2. Changes in physiological or clinical parameters.
3. Changes in knowledge and attitudes towards NCD risk factors.

### Secondary outcomes (outcomes reported along with health outcomes)

1. Change in utilisation of health promotion-related services (eg, smoking quitlines and counselling).
2. Costs/cost-effectiveness of the interventions.

### Search methods for identifying studies

Due to the diversity of population, settings in which interventions may be implemented and outcome types can be observed, a multistage electronic search strategy has been developed to identify relevant publications. A draft search strategy has been submitted as online supplementary file 2. Searches of published literature on effectiveness of NCD control interventions delivered through

targeted health promoting settings (focused on low/middle-income countries) will be done in the following biomedical, and general reference electronic databases, without restriction to language, however, the publication year is restricted to 2000–2018: PubMed (2000–2018), Excerpta Medica Database (2000–2018), OVID and WHO library. Cochrane Database of Systematic Reviews is another source of information that is proposed to be reviewed for relevant literature. The time period has been chosen considering the evidence that NCD prevention and control priorities have been set recently in low/middle-income countries primarily after WHO's call for global action. World Bank list (2018) will be used to define the countries or regions whose data will be used for effectiveness estimation.[23]

Further related studies will be identified by examining the reference lists of articles which have been already identified through above-mentioned database searches. In similar way, bibliographies of systematic and non-systematic review articles will also be examined to identify further related studies. Authors or trial investigators, for further information, will be contacted if any query pertaining to methodology, study outcome etc will arise. Abstracts and full text of identified manuscripts will be reviewed. Reference lists of systematic reviews obtained in the initial scoping reviews will be screened for relevant trials. Studies examining effectiveness and those evaluating cost-effectiveness will be reviewed separately.

### Searching other sources

The conference proceedings will be searched by hand wherever feasible (pending availability) for the last 5 years for conferences held in the field of NCDs, health promotion and health systems. In addition, trial registration website and WHO International Clinical Trials Search Portal[24] will be searched to identify trials in process or to compare the published trials with registered protocols. In addition trial registration websites of individual low/middle-income countries such as Chinese Clinical Trials Registry,[25] Clinical Trials Registry (India),[26] Clinical Research Information Service (Republic of Korea),[27] Pan African Clinical Trial Registry,[28] Sri Lanka Clinical Trials Registry,[29] Brazil[30] will be explored. We will also search different relevant web sources like Grey Literature Report (www.greylit.org) and Eldis (www.eldis.org) for relevant grey literature.

### Selection of trials and data abstraction

Two researchers (GJ/KK/PD/RP) will screen the titles and abstracts individually and in a manner independent of each other to identify their relevancy. Titles and the abstracts of articles identified using searches will be imported to EndNote.[31] All of the duplicates discovered in the process will be removed. In addition to assessment of every abstract and title by both researchers, the supervisor (JST) will further independently assess 50% of the abstracts. Retrieval of full-text articles will be done for articles for which the selection criteria is met

or declared unclear by both reviewers. If both reviewers agree to the situation of even one selection criteria not being met then full-text retrieval will not be done. For the titles/abstracts, where all selection criteria found categorised as 'no' by one reviewer were categorised entirely opposite or unclear by other reviewer, the final decision will be to retrieve the full text of the article for better review of selection criteria and if any disagreements are found during both stages then they will be resolved by discussion with third reviewer (JST). Studies with a control group other than routine care, studies with a sample size <30 and studies without a clearly defined intervention will be excluded. Articles written in a language other than English will be translated whenever possible. Multiple publications of the same study will be identified, grouped together and represented by a single reference.

### Data extraction and management

Data will be extracted using data abstraction form which will be developed on Microsoft Office Excel 2010. The data abstraction tool will be piloted on a random sample of five trials and modified as per the feedback from team. Data will be extracted by one reviewer (GJ) and checked against the original paper by a second independent reviewer (RP/KK/PD). Any disagreements during the process will be subject to discussion to reach a solution and if not resolved by discussion between the two then the issue will be resolved as per final decision of the third reviewer (JST). The data extraction for effectiveness assessment will be carried out in following domains: study description (authors, aims, objectives, ethical issues); trial details (details of location, population, intervention and control used); intervention details (allocation methods, theoretical basis, content of intervention, delivery, duration, targeted setting); outcome measures used and results with sufficient information; targeted risk factors, type of settings-based health promotion.

### Quality and risk of bias assessment in included studies

Risk of bias assessment will be done by using 'Cochrane risk of bias tool'.[32] Two authors will undertake assessment of risk of bias (GJ/PD/KK) and arbitration will be done by a third author in case of disagreements. (JST/MS/SP).

In order to assess the quality of a trial, Effective Public Health Practice Project Quality Assessment Tool (EPHPP) risk of bias tool will be used in addition to 'Cochrane risk of bias tool'.[33] Inter-rater reliability coefficients for exclusion as well as quality rating of the reviews will be reported.[34] The interpretation of findings will contain risk of bias assessment as risk of bias table. No restrictions will be made in the analysis on the basis of degree of risk of bias. For each primary outcome, a separate presentation of the risk of bias will be documented with respect to blinding and incomplete outcome assessment. Overall, strength of evidence assessed using EPHPP tool will be presented by inclusion in 'characteristics of included studies' table.

## Data analysis

Where data will allow, we will undertake meta-analysis. We will report the mean differences between change in intervention and control group with respect to quantitative data outcomes. In case measurement of same quantitative data outcomes has been undertaken in dissimilar manner, then we will report standardised mean differences between the intervention and control group. If we encounter a situation where we do not find a change per group, we will make use of end values where randomisation was successful. We will not include studies in meta-analysis in cases where there is non-availability of information on change per group and if there is reasonable risk of selection bias. For dichotomous outcomes, relative risk will be reported. 95% CIs will be reported with all of the effect estimates.

If it is unclear whether given study is a cRCT or not, the original investigator of the study will be contacted for further information. In case a situation is encountered where the original investigators have not included the clustering effect in their analyses, we will use standard methods as suggested by Cochrane group. First, we will request for provision of data for individual participants, so that an intracluster correlation coefficient (ICC) can be calculated, and the data can be reanalysed. In case we cannot get primary data, we will try to search the literature for appropriate ICC and make adjustments in the sample size in accordance with it. Correct effect estimates and SEs from cRCTs will be pooled using generic inverse-variance method.[35] Results will be marked as reanalysed if the data are reanalysed. If reanalysis would not be possible, it will be mentioned clearly.

In cases where measurement of outcomes was done in participants at more than one point of time, grouping of outcomes will be undertaken for similar time points. The minimum duration for which trials must have been implemented for extraction of outcomes is 3 months.

For the interventions with multiple comparison groups, all groups that meet the inclusion criteria will be included in the review. In trials with more than two comparisons, the relevant experimental and control groups will be combined to make a single pairwise comparison. If this combining of the groups is not possible, we will make multiple pairwise comparisons by dividing the sample size of the shared intervention group evenly across the comparison groups to avoid double counting of participants.

In cases where we find that data are lacking in clarity or there are missed values with respect to study methodology, participants lost to follow-up, outcome data or statistical techniques primary author followed by corresponding author of the study will be contacted through email. We will keep a condition of three reminders for requesting missing data from the authors. In case we do not obtain a response after these, outcome will not be included for the purpose of analysis. All of the data for missing outcomes will be reported in data extraction form and in the risk of bias table.

## Assessment of reporting biases

If there are more than 10 trials for an outcome, the likelihood of reporting bias will be explored by using funnel plots. A visual assessment of the funnel plots will be done for sources of asymmetry such as small-study effects, publication bias or others. If small-study effects are found to result in asymmetry then further sensitivity analysis will be undertaken to show its effects on the pooled results.

## Data synthesis

Detailed narrative synthesis of the results will be done for different settings and risk factors addressed. We will undertake separate meta-analyses with respect to each outcome. In order to incorporate any existing heterogeneity, the random-effects model will be used. A forest plot will be formulated for each comparison. However, the results of different outcomes under the trials will be pooled only if we find minimum of two studies and the studies are sufficiently homogeneous ($I^2$ statistic<75%). If either of the above-mentioned two criteria is not met, synthesis of results will be done only in a narrative manner. We will undertake meta-analysis using RevMan. STATA software will also be used for analysing the data for publication bias because the publication bias tests such as Egger's test are not applicable in RevMan.[36] For the primary outcomes of this review, a summary of findings table will be incorporated to take into account the number of participants and studies for each outcome, a summary of the intervention effect and a measure of the quality of evidence for each outcome according to Grading of Recommendations Assessment, Development and Evaluation considerations.[37]

## Subgroup analysis and investigation of heterogeneity

Outcomes of interventions identified under the review will be grouped by risk factors/diseases and type of setting. Such types of analyses will permit exploration of clinical as well as methodological heterogeneity between studies.[38] In order to make a comparison between different subgroups, we will conduct a standard heterogeneity test in RevMan 2012. This will be done by determining the $I^2$ statistic. Assessment of heterogeneity or the variability among the studies included in a meta-analysis will be done using visual inspection of overlap of CIs as well as by assessing statistical heterogeneity with the $\chi^2$ test. For $I^2$ statistic above 75% (an indication of substantial heterogeneity) results will not be pooled. We plan to undertake sensitivity analysis to make an assessment of how study size, study quality affect the review findings.

## Ethics and dissemination

No ethical issues are foreseen. Dissemination will be done by submitting scientific articles to academic peer-reviewed journals. We will present the results at relevant conferences and meetings.

## DISCUSSION

Sustained support in international health promotion research has led to generation of vast volume of literature

on settings-based approach. Through this review, we aim to explore their success in NCD prevention. This will help to conclude and generalise the targeted interventions for NCDs in different settings among low/middle-income countries, thus, ultimately guiding the resource allocation towards preventive and curative services using settings-based approach.

**Contributors** Concept and design of study: JST, GJ, SP and MS. Refining the methodology and data abstraction tools: GJ, KK, PD and RP. Drafting the manuscript: GJ, KK, PD and RP. Revising the manuscript critically for important intellectual content: MS, SP and JST. Final approval of the version to be published: JST, SP and MS.

**Funding** This research received no specific grant from any funding agency in the public, commercial or not-for-profit sectors.

**Competing interests** None declared.

**Patient consent** Not required.

**Ethics approval** The Institute Ethics Committee of the Post Graduate Institute of Medical Education and Research approved the doctoral research protocol under which this review is being done.

**Provenance and peer review** Not commissioned; externally peer reviewed.

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
