## [Reviewer comments · BMJ Open]

ARTICLE DETAILS

TITLE (PROVISIONAL)	Effectiveness of Targeting the Health Promotion Settings for Non-Communicable Disease Control in Developing Countries: Systematic Review Protocol
AUTHORS	Jeet, Gursimer; Thakur, JS; Prinja, Shankar; Singh, Meenu; Paika, Ronika; Kunjan, Kunjan; Dhadwal, Priya

VERSION 1 - REVIEW

REVIEWER	Dr.M.Meghachandra Singh Director-Professor Department of Community Medicine Maulana Azad Medical College New Delhi 110002 India
REVIEW RETURNED	13-Oct-2016

GENERAL COMMENTS	Some corrections are needed before finalization of the protocol. Details of such corrections to be done are shown in the text of the protocol at relevant places. Short form "NCD" in the title should be expanded. Use of search engine should have also included words such as "county" in the settings since in few LMICs it is used such as China, "community trial in NCD/diabetes/cardiovascular diseases/stroke/cancer" might have yielded some more studies. Requires changes in language at places indicated in the text. References need to be rewritten as per Vancouver style as indicated in several places e.g. use et al after six authors, avoid use of full stop after initial of name, add [cited on....], available from ... The reviewer also provided a marked copy with additional comments. Please contact the publisher for full details.
---

REVIEWER	Susan Horton University of Waterloo, Canada
REVIEW RETURNED	05-Dec-2016

GENERAL COMMENTS	This is an ambitious review. The text needs significant clean-up to improve clarity of exposition. I think it is worth thinking a bit more about the methodology used. Although individual-based, double-blind RCTs are the gold standard for studies of individuals, once trials are no longer double-blind and also clustered (such that participants in one school are the intervention group, and those in a different schools are not) then it becomes more important to control for differences in characteristics of children attending different schools. Similarly, if a village is the unit of randomization, then there
---

is the possibility of effects from the intervention bleeding over into the control. In these circumstances, the differences between an observational trial (with control) and a cluster-randomized trial, becomes less clear. I explain this because in a previous literature review I have undertaken on interventions for smoking and obesity in schools, the majority of studies were observational with control, and the cluster RCTs are not representative of the literature. I would recommend at minimum including (for perhaps a restricted range of interventions) comparing both literatures in terms of numbers of studies. Otherwise the risk is that the paper uses a very well-specified methodology and an approach which seems very careful (bias tools, forest plots, etc) but only covers a biased set of the literature.

I think also some greater clarity is needed on what settings are being included. Figure 1 includes some things as settings that are not mentioned much or at all in the text: “the market” is not really a setting; “home” is also not really a setting as used elsewhere in the document – individuals in homes are often reached either through mass media or one-on-one. It makes little sense to include “old age” as a setting if only death up to age 69 is included. A figure (or table) providing examples of the kinds of interventions expected to be covered would be useful.

Specific comments

Methods: is there a reason for not including Scopus? My experience is that studies of programs for adolescents in school (and possibly others) are not only in the medical journals. It is possible that articles in Scopus are less likely to be RCTs perhaps, but see general comments on this.

P5, line 28: the usage “settings-based approaches” is grammatically correct. I would encourage consistency throughout. I.e. replace “setting based approaches” (line 38, same page), line 10 page 7 “setting based approach”, and other similar instances.

P5, line 51: the authors explain that effects are likely to percolate from one setting to another. I agree – so how does this make the RCT method remain valid, but observational trials are not?

P5, line 56: “be brought to a de-escalating trend” is very cumbersome. Are the authors trying to say “slow down the rate of increase” or something?

P6, line 21: “a Supplementary File” not “Supplementary File”. Also same line “design, implementation and sustainability” or “designing, implementing and sustaining”

P6, line 34: islands, prisons (not island, prison) to be consistent.

P6, line 34: if population is only being considered up to age 69 years, then does it make sense to cover old age homes? These are unlikely to report separate results for interventions on those below 70 and above 70.

P6, line 37: define “central” schools (this isn’t a commonly-used word in UK/US English).

P6, line 48: defined “aided” institutions – same as previous comment

	P7, line 6: “becomes” not “become” P7, line 14: stress does not appear in Figure 1. “Chronic disease management” does not appear elsewhere in the paper and doesn’t seem to be the same as health promotion. Health promotion can be used to reduce risky behavior (traffic accidents, helmet use) so the paper has to rule out injury. Risky sexual behavior also leads to chronic disease but be clear that is being excluded. Use of controlled substances also affects chronic disease but these are not being included. It has to be clear what is included and what is not, and consistency through the paper is important. If the paper is about four types of risk, then stick to those four types throughout. P7, line 17-24: the discussion of the type of interventions to be included is quite limited. It would be useful to give examples beyond this brief discussion. The type of interventions isn’t given in Figure 1, nor included in the search terms (because it is not being limited) – but it would be valuable to give examples of what is expected either in a table or a figure. For example internet-based programs are likely another type not mentioned, probably one-on-one counselling, etc. P7, line 56: capitalize Bank (World Bank, not World bank). And specify that all low and middle income countries will be included (since the World Bank also provides data on high income countries). Will non-OECD but high income countries (Saudi Arabia etc) be included or excluded? P8, line 21: is a list of conferences to be searched being provided? P8, lines 49-50: “Studies with a control group other than routine care, studies with a sample size <30 and studies without a clearly defined intervention will be excluded” (grammar updated). P10, line 3: instead of “we will be” use “it is” P10, line 28: “dividing” not “divide” P10, line 34, do not capitalize “primary” as in “primary author” P10, line 38: “in case we do not obtain a response” instead of “in case we do not a response..” P10, line 46: “using” (typo) P11, line 4: delete “will use” P11, line 7/8: insert “the studies are” before “sufficiently homogeneous”. Also “if either of the above-mentioned...” not “if any...” P11, line 43: delete “the” in “on the all four common risk...” P11, line 47: what is a “target-led” intervention? Why is there now a discussion of targeting being introduced here? Figure 1, p16. No legend is provided for the Figure, nor a source. I find the different levels useful, but not all the arrows (everything in the row above seems to affect everything in the next row anyway). In terms of settings, “markets” are not mentioned in the text. Homes are mentioned. But in what ways are homes and markets settings
--	--

	like work and school based interventions? I would assume one might use population-level policies or mass media to reach people in their homes. But is this really a settings-based study? Religious institutions are included in the Figure – if so, should be reflected in the text. I have previously explained that I don't think it makes sense to include "old age" in the survey. I recommend redrawing the Figure.
--	---

VERSION 1 – AUTHOR RESPONSE

Reviewer: 1

Comment 11: Short form "NCD" in the title should be expanded.

Response 11: The abbreviation has been expanded as "Non-Communicable disease"

Comment 12: Use of search engine should have also included words such as "county" in the settings since in few LMICs it is used such as China. "Community trial in NCD/diabetes/cardiovascular diseases/stroke/cancer" might have yielded some more studies.

Response 12:

a. The search term county has been added to the search strategy

b. "community trial in NCD/diabetes/cardiovascular diseases/stroke/cancer" has been added to the search strategy

Comment 13: Requires changes in language at places indicated in the text.

Response 13: The suggested changes have been completed

Comment 14: References need to be rewritten as per Vancouver style as indicated in several places e.g. use et al after six authors, avoid use of full stop after initial of name, add [cited on....], available from ...

Response 14: The references have been rewritten as suggested.

Reviewer: 2

Comment 15: Methods: is there a reason for not including Scopus? My experience is that studies of programs for adolescents in school (and possibly others) are not only in the medical journals. It is possible that articles in Scopus are less likely to be RCTs perhaps, but see general comments on this.

Response 15: Scopus is a good search engine and we had thought of including it in our search but as per previous experience of authors (and reviewer as well), it was agreed upon as a good source of literature on programs and covering Ovid, Medline comprehensively covers the literature. In purview of our current focus we have decided to include it.

Comment 16: P5, line 28: the usage "settings-based approaches" is grammatically correct. I would encourage consistency throughout. i.e. replace "setting based approaches" (line 38, same page), line 10 page 7 "setting based approach", and other similar instances.

Response 16: The suggested change has been made.

Comment 17: P5, line 51: the authors explain that effects are likely to percolate from one setting to another. I agree – so how does this make the RCT method remain valid, but observational trials are not?

Response 17: From percolation of effects of an intervention into surroundings settings, authors refer to different setting enclosing the main setting being targeted. Percolation of positive or negative effects into the setting may affect the overall effectiveness of trial. For this, the trial design should be robust enough to capture any major influence. We did not mean here percolation of effect into control arm. This aspect becomes difficult to control in community settings, due to which such trials become difficult. However it is not impossible to achieve.

Comment 18: P5, line 56: "be brought to a de-escalating trend" is very cumbersome. Are the authors trying to say "slow down the rate of increase" or something?

Response 18: The language has been simplified.

Comment 19: P6, line 21: "a Supplementary File" not "Supplementary File". Also same line "design,

implementation and sustainability” or “designing, implementing and sustaining”

Response 19: The corrections have been made.

Comment 20: P6, line 34: islands, prisons (not island, prison) to be consistent.

Response 20: The corrections have been made and manuscript has been checked for any other discrepancies.

Comment 21: P6, line 34: if population is only being considered up to age 69 years, then does it make sense to cover old age homes? These are unlikely to report separate results for interventions on those below 70 and above 70.

Response 21: The purview of interventions under our review includes settings based approach for delivery of health promotion interventions. According to the WHO definition of settings, old age homes fall under the category of entities in which people engage in daily activities wherein environmental, organizational, and personal factors may interact to affect health and wellbeing. We would like to explore the effects of targeted NCD control interventions in the settings of old age homes with people above 70 years of age as they may possess different characteristics compared to those in other age groups with respect to physical activity, going to market etc. We do not intend to limit the age group to 0-69 years and we are considering old age homes as sub population. The confusion is arising due to a stratification attempted by us in Figure 1. Figure has been revised for better clarity.

Comment 22: P6, line 37: define “central” schools (this isn’t a commonly-used word in UK/US English).

Response 22: The definitions have been added to the manuscript.

Comment 23: P6, line 48: defined “aided” institutions – same as previous comment

Response 23: The definitions have been added to the manuscript.

Comment 14: P7, line 6: “becomes” not “become”

Response 14: The correction has been made.

Comment 15: P7, line 14: stress does not appear in Figure 1. “Chronic disease management” does not appear elsewhere in the paper and doesn’t seem to be the same as health promotion. Health promotion can be used to reduce risky behavior (traffic accidents, helmet use) so the paper has to rule out injury. Risky sexual behavior also leads to chronic disease but be clear that is being excluded. Use of controlled substances also affects chronic disease but these are not being included. It has to be clear what is included and what is not, and consistency through the paper is important. If the paper is about four types of risk, then stick to those four types throughout.

Response 15: Manuscript has been revised and the risk factors to be included have been mentioned with more clarity.

Comment 16: P7, line 17-24: the discussion of the type of interventions to be included is quite limited. It would be useful to give examples beyond this brief discussion. The type of interventions isn’t given in Figure 1, nor included in the search terms (because it is not being limited) – but it would be valuable to give examples of what is expected either in a table or a figure. For example internet-based programs are likely another type not mentioned, probably one-on-one counselling, etc.

Response 16: Following the reviewer’s suggestion, we have added the details. (Page 7)

Comment 17: P7, line 56: capitalize Bank (World Bank, not World bank). And specify that all low and middle income countries will be included (since the World Bank also provides data on high income countries). Will non-OECD but high income countries (Saudi Arabia etc) be included or excluded?

Response 17: The suggested correction has been made. We will strictly follow the World Bank list of low and middle income countries.

Comment 18: P8, line 21: is a list of conferences to be searched being provided?

Response 18: We do not plan to enclose any list of conferences at protocol stage. However we will definitely add the details in final manuscript.

Comment 19: P8, lines 49-50: “Studies with a control group other than routine care, studies with a sample size <30 and studies without a clearly defined intervention will be excluded” (grammar updated).

Response 19: The correction has been made.

Comment 20: P10, line 3: instead of “we will be” use “it is”

Response 20: The correction has been made.

Comment 21: P10, line 28: “dividing” not “divide”

Response 21: The correction has been made.

Comment 22: P10, line 34, do not capitalize “primary” as in “primary author”

Response 22: The correction has been made.

Comment 23: P10, line 38: “in case we do not obtain a response” instead of “in case we do not a response..”

Response 23: The correction has been made.

Comment 24: P10, line 46: “using” (typo)

Response 24: The correction has been made.

Comment 25: P11, line 4: delete “will use”

Response 25: The terms “will use” has been deleted

Comment 26: P11, line 7/8: insert “the studies are” before “sufficiently homogeneous”. Also “if either of the above-mentioned...” not “if any...”

Response 26: Both the corrections have been made to the manuscript.

Comment 27: P11, line 43: delete “the” in “on the all four common risk...”

Response 27: The suggested correction has been made to the manuscript.

Comment 28: P11, line 47: what is a “target-led” intervention? Why is there now a discussion of targeting being introduced here?

Response 28: The section has been redrafted.

Comment 29: Figure 1, p16. No legend is provided for the Figure, nor a source. I find the different levels useful, but not all the arrows (everything in the row above seems to affect everything in the next row anyway). In terms of settings, “markets” are not mentioned in the text. Homes are mentioned. But in what ways are homes and markets settings like work and school based interventions? I would assume one might use population-level policies or mass media to reach people in their homes. But is this really a settings-based study? Religious institutions are included in the Figure – if so, should be reflected in the text. I have previously explained that I don’t think it makes sense to include “old age” in the survey. I recommend redrawing the Figure.

Response 29: The figure has been redrawn for relevant changes. In addition we have provided text below figure to provide clarity to readers.

Comment 30: Modify the sentence to clearly state, “what is the comparison between developed and developing countries?”

Response: The sentence has been suitably modified.

Comment 31: P5: Change the word Kind to Kinds

Response: Correction has been made.

Comment 32: P6: The word set and “,” can be deleted

Response 32: Correction has been made.

Comment 33: P7: Add “Changes in” before ‘physiological’

Response 33: Correction has been made.

Comment 34: P8: Add “ever” after where

Response 34: Correction has been made.

Comment 35: P8 Final decision is not by retrieving full article, but it is the means to achieve final decision by a third investigator. Hence, change the wording appropriately.

Response 35: Suggested change has been made.

Comment 36: P9: Add here, "if not resolved by discussion between the two, then specify how to decide."

Response 36: Suggested change has been made.

Comment 37: Meta-analysis itself will be altogether a separate design than the present study protocol. It is better to exclude it from the present protocol.

Response 37: After discussions within review group, it was decided to keep the details pertaining to meta-analysis.

Comment 38: Several typo errors on page 10, need correction

Response 38: All the corrections suggested as comments in PDF have been made

Comment 39: P11: reword the phrase

Response 39: Correction has been made

Comment 40: P11: Can STATA give authentic data on publication bias

Response 40: For our other reviews, we rely on STATA to run tests specific to detection of publication bias.

VERSION 2 – REVIEW

REVIEWER	Dr.M.Meghachandra Singh Maulana Azad Medical College, New Delhi, India
REVIEW RETURNED	05-Mar-2017

GENERAL COMMENTS	The reviewer also provided a marked copy with additional comments. Please contact the publisher for full details.
---

REVIEWER	Susan Horton University of Waterloo, Canada
REVIEW RETURNED	25-Feb-2017

GENERAL COMMENTS	Although the authors state that the article has been proofread, I would suggest correcting the following grammatical mistakes (and perhaps not using this editor again when submitting any papers from the study): Line 73: insert "country": "developed country settings" Line 100: "developing countries. A....." instead of "developing countries, a" Line 111: "is more alarming" not "is alarming" Line 112: giving a percentage of the NCD deaths which are premature would allow people to get more alarmed, rather than simply stating "most" Line 121: "integrate" not "integrates" Line 123-124: should be written "suitable infrastructure, physical environment and health-promoting policies..." or "suitable infrastructure and physical environment and health-promoting policies.." depending what is meant (former seems preferable to me). Line 126: I believe "globalisation- and urbanisation-led..." is what is meant Line 133: "channeled" not "channelized" (if American spelling is being used) or I believe "channelled" is the British spelling Line 140: "be slowed" not "slowed" Line 163-4: "municipalities/communities" to be consistent with rest of sentence Line 176: I would recommend spelling "organized" and "unorganised" either both with z or with s Line 187: "settings-based approaches" not "settings based"
---

	approaches” and check full document that this has been written consistently throughout Line 197: “but is not limited to” not “but not limited to” Line 238: “wherever” is usually written as one word Line 248/9: not really clear what this sentence means. Is it that “..... and, in a manner independent of each other, identify the titles and abstracts for relevancy”? (i.e. the two readers are working independently?). Or is it that title and abstract independent of each other will be reviewed for relevancy, if so please rewrite sentence so this is clear. Line 294-5: do the authors mean “in cases where...” or “in cases in which..” rather than “in case”? Line 338: Do the authors mean: “We will use STATA software.....publication bias test such as Eggers test...”? Line 364: suggest “countries, thus ultimately guiding” instead of “countries. Thus ultimately guiding..” such that the discussion ends with a full sentence.
--	--

VERSION 2 – AUTHOR RESPONSE

Reviewer 1:

Comment 2: Some minor corrections needed as highlighted by comment dialogue boxes at places. Corrections also required in the references section as given in the dialogue boxes.

Response 2: The corrections suggested in dialogue boxes have been made throughout the document including references section.

Reviewer 2:

Comment 3: Line 73: insert “country”: “developed country settings”

Response 3: The suggested change has been made.

Comment 4: Line 100: “developing countries. A.....” instead of “developing countries, a”

Response 4: The suggested change has been made.

Comment 5: Line 111: “is more alarming” not “is alarming”

Response 5: The suggested change has been made.

Comment 6: Line 112: giving a percentage of the NCD deaths which are premature would allow people to get more alarmed, rather than simply stating “most”

Response 6: The suggested change has been made as “The figure is alarming than it appears as 52% of the NCD deaths that occurred were premature, i.e. among people aged 30-70 years”

Comment 7: Line 121: “integrate” not “integrates”

Response 7: The suggested change has been made.

Comment 8: Line 123-124: should be written “suitable infrastructure, physical environment and health-promoting policies...” or “suitable infrastructure and physical environment and health-promoting policies..” depending what is meant (former seems preferable to me).

Response 8: The suggested change has been made as “It is also indicated that support from staff, suitable infrastructure, physical environment and health promoting policies have potential to positively influence the health of a person especially in these kinds of settings”

Comment 9: Line 126: I believe “globalisation- and urbanisation-led...” is what is meant

Response 9: The suggested change has been done as “In order to decrease the burden of proximal and distal determinants of NCDs which has increased due to globalization and urbanization, settings-based approach may be implemented in a complete and inclusive manner by targeting singular or multiple settings simultaneously”

Comment 10: Line 133: “channeled” not “channelized” (if American spelling is being used) or I believe “channelled” is the British spelling

Response 10: The suggested change has been done and the word “channelled” has been added instead of “channelized”.

Comment 11: Line 140: “be slowed” not “slowed”

Response 11: The suggested change has been made

Comment 12: Line 163-4: “municipalities/communities” to be consistent with rest of sentence

Response 12: The suggested change has been made

Comment 13: Line 176: I would recommend spelling “organized” and “unorganised” either both with z or with s

Response 13: The suggested change has been made and “s” has been used uniformly.

Comment 14: Line 187: “settings-based approaches” not “settings based approaches” and check full document that this has been written consistently throughout

Response 14: The suggested change has been made in the whole document using “settings-based” instead of “settings based”.

Comment 15: Line 197: “but is not limited to” not “but not limited to”

Response 15: The suggested change has been made and “but is not limited to” has been added.

Comment 16: Line 238: “wherever” is usually written as one word

Response 16: The change has been made

Comment 17: Line 248/9: not really clear what this sentence means. Is it that “..... and, in a manner independent of each other, identify the titles and abstracts for relevancy”? (i.e. the two readers are working independently?). Or is it that title and abstract independent of each other will be reviewed for relevancy, if so please rewrite sentence so this is clear.

Response 17: This line states that the two researchers will be independently reviewing the titles and abstracts of the articles for relevancy (inclusion criteria). The sentence has been modified to provide clarity.

Comment 18: Line 294-5: do the authors mean “in cases where...” or “in cases in which..” rather than “in case”?

Response 18: The change has been made as “We will not include studies in meta-analysis in cases where there is non-availability of information on change per group....”

Comment 19: Line 338: Do the authors mean: “We will use STATA software.....publication bias test such as Eggers test...”?

Response 19: We have edited the sentence as “STATA software will also be used for analyzing the data for publication bias because the publication bias tests such as Eggers tests are not applicable in RevMan.

Comment 20: Line 364: suggest “countries, thus ultimately guiding” instead of “countries. Thus ultimately guiding..” such that the discussion ends with a full sentence.

Response 20: The suggested change has been made.

VERSION 3 – REVIEW

REVIEWER	Mongjam Meghachandra Singh Maulana Azad Medical College New Delhi India
REVIEW RETURNED	26-Apr-2017

GENERAL COMMENTS	Corrections incorporated
--------------------------

REVIEWER	Susan Horton University of Waterloo, Canada
REVIEW RETURNED	18-Apr-2017

GENERAL COMMENTS	no further comments
---------------------